# The Relationship between Obesity and Clinical Outcomes in Young People with Duchenne Muscular Dystrophy

**DOI:** 10.3390/nu14163304

**Published:** 2022-08-12

**Authors:** Natassja Billich, Justine Adams, Kate Carroll, Helen Truby, Maureen Evans, Monique M. Ryan, Zoe E. Davidson

**Affiliations:** 1Department of Nutrition, Dietetics and Food, School of Clinical Sciences at Monash Health, Faculty of Medicine, Nursing and Health Sciences, Monash University Melbourne, Victoria 3168, Australia; 2Neurology Department, The Royal Children’s Hospital Melbourne, Victoria 3052, Australia; 3School of Human Movement and Nutrition Sciences, The University of Queensland Brisbane, Queensland 4072, Australia; 4Murdoch Children’s Research Institute Melbourne, Victoria 3052, Australia; 5Department of Physiotherapy, School of Primary and Allied Health Care, Faculty of Medicine, Nursing and Health Sciences, Monash University Melbourne, Victoria 3168, Australia; 6School of Primary and Allied Health Care, Faculty of Medicine, Nursing and Health Sciences, Monash University Melbourne, Victoria 3168, Australia; 7Metabolic Medicine, The Royal Children’s Hospital Melbourne, Victoria 3052, Australia; 8Department of Paediatrics, Faculty of Medicine, Dentistry and Health Sciences, Melbourne University, Victoria 3010, Australia

**Keywords:** Duchenne muscular dystrophy, obesity, clinical outcomes, physical function, obstructive sleep apnoea, fractures

## Abstract

Background: Duchenne muscular dystrophy (DMD) is a severe X-linked neuromuscular disorder. Young people with DMD have high rates of obesity. There is emerging evidence that a higher BMI may negatively affect clinical outcomes in DMD. This study aimed to explore the relationship between obesity and clinical outcomes in DMD. Methods: This was a retrospective clinical audit of young people (two–21 years) with DMD. Height and weight were collected to calculate BMI z-scores to classify obesity, overweight and no overweight or obesity (reference category). Cox proportional hazards models determined the impact of obesity at five to nine years on clinical milestones including time to: loss of ambulation, timed function test cut-offs, obstructive sleep apnoea (OSA) diagnosis and first fracture. Results: 158 young people with DMD were included; most (89.9%) were steroid-treated. Mean follow-up was 8.7 ± 4.7 years. Obesity prevalence increased from age five (16.7%) to 11 years (50.6%). Boys with obesity at nine years sustained a fracture earlier (hazard ratio, HR: 2.050; 95% CI: 1.038–4.046). Boys with obesity at six to nine years were diagnosed with OSA earlier (e.g., obesity nine years HR: 2.883; 95% CI: 1.481–5.612). Obesity at eight years was associated with a 10 m walk/run in 7–10 s occurring at an older age (HR: 0.428; 95% CI: 0.207–0.887), but did not impact other physical function milestones. Conclusions: Although 50% of boys with DMD developed early obesity, the impact of obesity on physical function remains unclear. Obesity puts boys with DMD at risk of OSA and fractures at a younger age. Early weight management interventions are therefore important.

## 1. Introduction

Duchenne muscular dystrophy (DMD) is a severe X-linked paediatric neuromuscular disorder which affects approximately between 15.9 and 19.6 out of 100,000 live male births [1,2,3,4]. Mutations in the DMD gene result in the absence of dystrophin and lead to damaged muscles during contraction, replacement of atrophied muscles with fibrotic and adipose tissue, and progressive loss of physical function [5,6,7]. Long-term corticosteroid treatment is the mainstay therapy for DMD, with benefits including prolongation of ability to independently ambulate from 10 years in steroid-naïve boys to 13 years in steroid-treated boys [8,9].

Short stature and susceptibility to both over- and underweight are well documented in DMD [10,11]. Body mass index (BMI) is typically higher in both steroid-treated and steroid-naïve boys with DMD compared to typically developing children [12,13]. One Australian study which explored longitudinal BMI patterns identified a persistently high BMI from three to 12 years of age compared to typically developing reference populations [12,14]. The prevalence of obesity peaked at 10 years of age, at which time 48% of young people with DMD had obesity [12]. While steroids can cause weight gain [8], even in steroid-naïve cohorts the rate of obesity is up to 70% [15,16,17]. In the late non-ambulatory phase, young men with DMD are at risk of underweight with causes including dysphagia and feeding difficulties, progressive muscle wasting with age and a potential hypermetabolic resting energy expenditure [12,18,19]. There is emerging evidence that a higher BMI and higher fat mass are associated with worsened physical function, sleep, lung function, and metabolic risk factors in DMD [20,21,22,23,24,25,26,27,28]. Most studies have explored the impact of BMI on clinical outcomes in secondary analyses or have analysed only one group of clinical outcomes (e.g., lung function). No studies have comprehensively looked at the effect of BMI status on multiple clinical outcomes or have studied the impact of BMI on the time taken for clinically relevant milestones—such as age at loss of ambulation—to occur. To address this gap, this study aimed to investigate the impact of BMI status on clinically relevant milestones in young people with DMD. To conduct this investigation, this study also addressed changes in anthropometric measures over time in DMD. An additional secondary aim was to identify factors predicting obesity in DMD.

## 2. Materials and Methods

### 2.1. Study Design and Setting

This was a retrospective clinical audit of patients attending the Neuromuscular Clinic at the Royal Children’s Hospital (RCH) in Melbourne. The Neuromuscular Clinic at RCH is a multi-disciplinary clinic specialising in the management of paediatric patients with a range of neuromuscular conditions including DMD. RCH hosts the largest single neuromuscular clinic in Australia and provides a service to all young people with neuromuscular conditions from Victoria and Tasmania.

### 2.2. Population

Eligible medical records were identified from central clinic patient lists. Eligible patients had a diagnosis of DMD and attended the neuromuscular clinic at RCH between January 2011 and March 2018. Diagnosis was made by either genetic testing or muscle biopsy confirming DMD. The cut-off date (2011) reflects the period where scanned and electronic medical records were introduced in the hospital. The electronic medical record system at RCH contains all progress notes, investigations, pathology results, scans, growth data and correspondence for each patient. Patient records were excluded if anthropometry and the majority of clinical information was missing e.g., attended one appointment but then moved. Patients with Becker muscular dystrophy were excluded. There were no age limits set for patients. As data were collected from a paediatric clinic most subjects were below 18 years at last follow up, however there were some older patients (maximum 21 years).

### 2.3. Procedures

Data from medical records were collected retrospectively and managed using REDCap (Research Electronic Data Capture) [29]. REDCap is a secure, web-based software platform designed to support data capture for research studies. The REDCap database was built and piloted amongst three investigators (NB, KC, ZD). The database functionality and data collection accuracy were piloted by two investigators extracting test data from the electronic medical record of one patient. Extracted data was then compared and data collection guidelines were developed to ensure consistency. Fixed database responses and number limits were used where possible to optimise data accuracy. Medical records were reviewed, and data was collected by one researcher (NB, JA or KC). Uncertainty about the interpretation of data within a record was resolved by consensus among the researchers. For eligible patients, data were collected from the date of the first neurologist appointment at RCH or date of diagnosis, whichever came first. The last data point was from 31 July 2018—the date data collection commenced—or earlier if the patient was ‘non-active’ in the clinic. Patients were non-active if they had transitioned to adult services, were lost to follow-up, moved, attended an alternative clinic, or died.

### 2.4. Outcomes

#### 2.4.1. Clinical Characteristics

Clinical characteristics collected from medical records included age at first neurologist appointment, length of follow up, diagnostic information, treatment with corticosteroids and the presence of co-morbidities (neurodevelopmental disabilities, mental health conditions or respiratory complications). Date of diagnosis was the date of genetic or muscle biopsy confirmation of DMD. Genetic mutation was categorised based on type of mutation (deletion, duplication or duplication/triplication, point mutation) and the dystrophin isoforms maintained based on the location of the mutation [30]. “Steroid-treated” was defined as any period of daily or intermittent dosing of oral corticosteroid therapy (prednisolone, deflazacort or other). In Australia, prednisolone is prescribed as first line therapy while deflazacort can be accessed via the Special Access Scheme if significant side effects such as weight gain and behaviour issues occur with prednisolone. Length of follow up was defined as the time from initial appointment with the neurologist until the last data point prior to July 2018 or until the patient was non-active in the clinic (whichever came first). Whether the young person saw a dietitian through the neuromuscular clinic was also noted.

#### 2.4.2. Anthropometric Data

Height and weight data were collected from clinic notes, growth charts, lung function test (spirometry), echocardiography and dual-energy X-ray absorptiometry (DXA) scan reports. In clinical practice for non-ambulatory patients, height is estimated by imputing a proxy measure (e.g., ulnar length) into an estimative equation. Estimated heights were also collected. Anthropometric data from birth to two years of age were excluded due to the limited data available. Height and weight measures were used to calculate BMI (weight kg/height m^2^) and height, weight and BMI z-scores were calculated using Cole’s LMS method [31]. BMI z-scores were used to classify patients into BMI status categories based on the Centers for Chronic Disease and Prevention (CDC) reference values: underweight, healthy weight, overweight and obesity [14].

#### 2.4.3. Clinical Milestones

Clinical milestones were defined as clinically relevant events in DMD disease progression e.g., age at loss of independent ambulation. Most clinical milestones were selected a priori from the literature (Table 1), with two exceptions; timed supine-to-stand > 7 s and 10 m walk/run in 7–10 s which were included as exploratory outcomes. The timed supine-to-stand > 7 s was included after observing that few patients (*n* = 4) reached a >30 s supine-to-stand, which is indicative of loss of ambulation in the proceeding months [32,33]. A timed 10 m walk/run in 7–10 s was included due to the clinical observation that this milestone typically indicates a period of rapid functional decline. For timed function tests (10 m walk/run, supine-to-stand and four stair climb), North Star Ambulatory Assessment (NSAA), six min walk distance (6 MWD) and forced vital capacity (FVC), only assessments performed above seven years were analysed, as younger boys are potentially still gaining motor skills and improving performance [32]. When exact dates were not known for clinical milestones (e.g., date at loss of ambulation or first fracture) the first day of the documented month was recorded.

### 2.5. Data Cleaning

Individual weight, height and BMI plots were reviewed for each subject by two Accredited Practising Dietitians (NB, ZD). Implausible or duplicate values were identified and removed by consensus, e.g., implausible values were decreases in height or a change in weight of 20 kg. Physical function and mobility outcome measures were reviewed by two physiotherapists (JA, KC).

### 2.6. Approvals and Patient Consents

Ethical approval was granted for this retrospective clinical audit by the Royal Children’s Hospital Research Governance Office (LNR/18/RCHM/233). As a clinical audit, this study was exempt from obtaining consent from patients. Reporting of this retrospective cohort study conforms to the STROBE statement [36].

### 2.7. Statistical Analysis

Data analysis was performed using SPSS statistical software (IBM Corp. Released 2019. IBM SPSS Statistics for Windows, Version 26.0. Armonk, NY: IBM Corp). Continuous parametric data is reported as mean and standard deviation (SD) and continuous non-parametric data is reported as median and interquartile range (IQR). To describe anthropometric measures over time, absolute values for height, weight and BMI were presented graphically and overlaid with CDC smoothed percentile data [37]. The closest BMI z-score value per year of age (two to 20 years) for each individual was categorised as underweight, healthy weight, overweight or obesity using the CDC reference values [37]. To determine the impact of BMI status on clinical outcomes, time-to-event analyses using a Cox proportional hazards models were conducted, and data were presented graphically using Kaplan-Meier curves. Events variables were clinical milestones and time variables were the age at which the milestone occurred. When the clinical milestone events did not occur, subjects were censored at the end of follow-up: July 2018 or the date at last neuromuscular clinic for those who had transitioned to adult services, died, moved, or lost to follow-up. If BMI or clinical milestone data was not available subjects were coded as missing. For each clinical milestone, five analyses were conducted for BMI status at age groups five through to nine years. The BMI status of each individual at each age was determined by the BMI measure closest to their birthday. BMI categories were no overweight or obesity (reference category); overweight; and obesity. Those who were underweight or a healthy weight were categorised as no overweight or obesity due to the low number who were underweight (*n* = 1 to 4). Some analyses were adjusted for covariates which were selected a priori: time to first fracture was adjusted for zoledronic acid treatment and time to scoliosis diagnosis was adjusted for ambulatory status. Some patients within the RCH neuromuscular clinic received prophylactic zoledronic acid treatment prior to their first fracture as part of a clinical trial [38]. Due to the known link between steroid treatment and fractures, time-to-event analyses for first fracture were also adjusted for steroid status (treated or naïve) prior to first fracture. Missing data for individual analysis are reported for each model. For Cox proportional hazard models, hazard ratios (HR) and 95% CI are reported.

To identify factors that predicted the likelihood of obesity, a generalized estimating equation (GEE) was used to account for within-patient repeated measures. The GEE was computed using an unstructured correlation matrix with a binary logistic distribution. The dependant variable was obesity (yes/no) and the independent variables were: age, length of follow-up, age at diagnosis, ambulation status (ambulant/non-ambulant), >7 s 10 m walk/run (yes/no), one fracture (yes/no), one dietitian visit (yes/no), scoliosis surgery (yes/no), steroid treatment (naïve/prednisolone/deflazacort/other), dystrophin isoform maintained and neurodevelopmental disability diagnosis (yes/no). Time-dependant variables were ambulation status, >7 s 10 m walk/run, one fracture, one dietitian visit, scoliosis surgery and steroid treatment. The time-dependency of these variables was considered in the coding of each BMI measure (obesity yes/no) over time. For this analysis, odds ratio (OR) and 95% confidence intervals (CI) are reported. For all analyses, a *p*-value of <0.05 was considered statistically significant.

## 3. Results

### 3.1. Clinical Characteristics

Of the young people with DMD identified, three were excluded as there was insufficient data available in medical records: one moved overseas, one was managed at an alternative hospital, and one was lost to follow up. One hundred and fifty-eight young people with DMD met the eligibility criteria for which clinical characteristics are described (see Table 2). Most were steroid-treated (*n* = 142, 89.9%). Fifty-seven boys were enrolled in a pharmaceutical trial during the follow-up period. There were 385 (13.4%) height and 370 (11.3%) weight that were implausible or duplicates and were excluded during the data cleaning process. One hundred and fifty-six subjects had anthropometric measures available; the total number of observations included were 2480 height, 2902 weight and 2456 BMI measurements.

### 3.2. Anthropometry across Age Groups

Absolute values for height, weight and BMI for young people with DMD were plotted on CDC percentile charts in Figure 1a–c. Boys with DMD exhibited slowed height growth compared to the CDC cohort. Height growth slowed from six years of age; by approximately 11–12 years most boys were below the third centile for height. Until approximately six to seven years, most weight measurements were within the normal limits of the percentile charts, after which the variance increased and values both over the 97th percentile and below the third percentile were observed in weight. For BMI, there was a wide spread of values that fell across the limits of the percentile charts, over the 97th percentile and below the third percentile, and variance increased with increasing age.

### 3.3. BMI Status

Obesity prevalence steadily increases from five years (16.7%) until it peaks at 11 years (50.6%) and then declines again to 25.0% at 19 years, see Figure 2. The highest prevalence of underweight was observed in those aged 18 years (20.0% underweight). After 18 years, available BMI measures reduced considerably (*n* = 12 for 19 years and *n* = 4 for 20 years) and no individuals in these age groups were underweight.

### 3.4. Impact of Obesity on Clinical Milestones

Descriptive data for clinical milestones can be found in Table 3. For each age group analysed (five to nine years), BMI measure available were: five years *n* = 78; six years *n* = 88; seven- and eight-years *n* = 97; and nine years *n* = 95. Data for all Cox proportional hazard model analyses are summarised in supplementary Appendix A.

#### 3.4.1. Obesity and Physical Function

BMI status at any age between five to nine years did not predict time any of the following clinical milestones: loss of ambulation, 10 m walk/run in >10 s, supine-to-stand in >7 s, four stair climb in >8 s, 6 MWD < 325 m or NSAA total score of 9 (see supplementary Appendix A). For the 10 m walk/run in 7–10 s, boys with obesity at eight years reached this milestone later (i.e., maintained a faster time for longer) compared to those without overweight or obesity (hazard ratio (HR): 0.428; 95% confidence interval (CI): 0.207–0.887), see supplementary Appendix A. When analysing BMI status at eight years, the median time to the 10 m walk/run in 7–10 s milestone was 11.8 years (95% CI 10.9–12.6) for those with obesity, 12.0 years (95% CI 10.2–13.7) for overweight and 10.6 (95% CI 10.3–10.9) for those without overweight or obesity (log rank *p*-value = 0.033). BMI status at other ages did not affect the time to a 10 m walk/run in 7–10 s.

#### 3.4.2. Obesity and First Fracture

In analyses unadjusted for zoledronic acid treatment, obesity at age six did not significantly impact time to first fracture (see supplementary Appendix A). However, when adjusting for zoledronic acid treatment (*n* = 19 treated prior to first fracture) those with obesity at six years were younger at first bone fracture compared to those without overweight or obesity (zoledronic acid HR: 2.327; 95% CI: 1.055–5.134), see Figure 3a. Obesity at six years was also associated with younger age at first fracture when adjusted for steroid status prior to first fracture (obesity HR: 2.196; 95% CI 1.019–4.733). Those with obesity at nine years also sustained a fracture earlier (see Figure 3b). This remained significant when adjusting for zoledronic acid treatment (obesity HR: 2.191; 95% CI: 1.094–4.387), when only including those who sustained a crush fracture (obesity HR: 2.106; 95% CI: 1.009–4.394, adjusted analysis), and when adjusting for steroid status (obesity HR: 2.169 95% CI 1.076–4.373). Median time to first fracture was: no overweight or obesity, 16.1 years (95% CI: 13.9–18.3); overweight, 14.7 years (95% CI: 14.1–15.4); and obesity, 12.3 years (95% CI: 11.3–13.3). Overall log rank *p*-value = 0.056. In unadjusted and adjusted analyses, obesity at five, seven and eight years did not impact time to first fracture (Appendix A).

#### 3.4.3. Obesity and Respiratory Function

Those with obesity at six to nine years were 2.2 to 3.4 times more likely to be diagnosed with OSA at any time point compared to those without overweight or obesity (see Figure 4a and supplementary Appendix A). Median time to OSA diagnosis was 13.6 years (95% CI: 10.6–16.6) for those with obesity; and 15.8 years (95% CI: 13.7–18.0) for those without obesity, overall log-rank *p*-value = 0.005. The Cox proportional hazards model indicated a higher likelihood of earlier diagnosis of OSA for those with obesity at age seven; however, the overall distribution across BMI status categories was not significantly different (log rank *p*-value = 0.130). OSA was diagnosed approximately two years earlier for boys with obesity at both eight (no overweight or obesity 15.8 years; 95% CI: 15.0–16.6 vs. obesity 13.6 years; 95% CI: 10.8–16.4, log rank *p*-value = 0.004) and nine years (no overweight or obesity 15.3 years; 95% CI: 14.3–16.2 vs. obesity 13.6 years; 95% CI: 10.8–16.4, log rank *p*-value < 0.001).

Boys with obesity at seven years commenced CPAP earlier than those without over-weight or obesity (obesity HR 3.735; 95% CI: 1.053–13.252, see supplementary Appendix A). Analyses at other ages were limited by small numbers of individuals initiated with CPAP (*n* = 14 to 24). Obesity did not impact time to a FVC < 1 L or scoliosis diagnosis (Appendix A).

### 3.5. Factors Associated with Obesity

GEE analysis determined factors associated with increased odds of obesity (see the supplementary Appendix A for the outcomes of the full model). There was a higher likelihood of obesity in boys referred to the clinic dietitian (OR 1.188 95% CI 1.064–1.327) and who were treated with prednisolone followed by deflazacort (OR 1.298 95% CI 1.093–1.543, reference category steroid-naïve). Factors associated with reduced odds of obesity was scoliosis surgery (OR 0.449 95% CI 0.351–0.574) and neurodevelopmental disability (OR 0.826 95% CI 0.695–0.983). To further understand the relationship between scoliosis surgery and BMI, post-hoc descriptive analysis of BMI z-score was conducted. For those who had scoliosis surgery during the follow up period (*n* = 15), the median pre-operative BMI z-score was 0.78 (IQR −0.33, 1.45 *n* = 166 BMI observations) and post-operative was −0.18 (IQR −2.09, 0.87, *n* = 111 BMI observations). Age, ambulatory status, completing a 10 m walk/run in 7 s and dystrophin isoforms maintained did not predict obesity.

## 4. Discussion

In this predominantly steroid-treated (90%) cohort, between one in ten and one in two young people with DMD had obesity. The rate of obesity amongst young people with DMD is between two to six times higher than the general population of Australian children and adolescents [39]. Young people with DMD and obesity are diagnosed with OSA and sustain an initial fracture earlier than their peers without overweight or obesity. It remains unclear what impact obesity has on physical function; obesity at age eight was associated with a later time to a slower (7–10 s) 10 m walk/run, however there was no impact of BMI status on any other measure of physical function.

Obesity at two time points, six and nine years, increased the likelihood of sustaining a fracture earlier. These significant findings were independent of steroid treatment in adjusted analyses. Analysis at other ages may have been limited by small sample sizes. Increased risk of fractures in children with obesity has also been observed in typically developing populations [40]. In DMD, both steroids and weight gain may lead to fractures in DMD. Steroid treatment can cause weight gain which may independently increase fracture risk [8], and steroids can also impact bone mineral density and lead to fractures [41]. Our sample size was too small to analyse differences between steroid types (e.g., deflazacort vs. prednisolone); however, in a recent randomised clinical trial there were no significant differences in rates of fractures between daily prednisolone and deflazacort [42]. In those with obesity and/or those taking steroids, there is a need for close surveillance of fracture risk factors such as vitamin D deficiency, low dietary calcium intake and falls. Dietitians and physiotherapists play an integral role in advising families on how to mitigate these risk factors to help prevent fractures, which causes a significant burden and can lead to loss of ambulation and decline in function and strength [43].

Diagnosis of OSA was up to two years earlier in those with obesity compared to those without overweight or obesity. Young people with DMD who have obesity may also require CPAP earlier, however our analysis exploring this was limited by small sample sizes. OSA and other sleep disturbances are prevalent in DMD and can have a significant impact on the quality of life and fatigue [27,44,45,46]. The relationship between obesity and OSA may be cyclical and poses challenges in the clinical management of both conditions: obesity causes OSA due to anatomical obstruction, while sleep disturbance causes fatigue and reduced perceived quality of life which may create barriers to weight management strategies (e.g., fatigue as a barrier to participating in physical activity). As well as the burden of OSA on individual patients and families, there are also significant economic implications due to the increased need for healthcare services [47,48]. We did not find an association between BMI status and time to FVC < 1 L, a milestone that negatively predicts survival in DMD [49]. Previous literature has shown that a lower BMI is associated with a lower FVC [12,21], however as we had few children in our sample that were underweight we were unable to observe this effect.

We observed little effect of obesity on time to clinical milestones related to physical function. Obesity at only one of the analysed age groups (eight years) was protective against a slower time (7–10 s) in a 10 m walk/run. We are unable to draw conclusions from this finding as this relationship was not observed for obesity at other age groups, nor for other functional outcome measures. Furthermore, the time cut-off of 7–10 s was included as an exploratory outcome which has not been used previously. A time cut-off of >10 s for the 10 m walk/run is used in prior literature and is a marker of loss of ambulation in the following 12 months [32,33]. Anecdotally, a time of 7–10 s is indicative of a period of rapid functional decline following this milestone. Body composition is a potential mediating factor in the relationship between obesity and a later time to a 7–10 s 10 m walk/run. Ambulant boys with a higher BMI may also have a higher muscle mass [50] and therefore may maintain strength, function and walking/running speed for longer. The response to steroids may also play a role in protecting against a slower walk/run but also causing a higher BMI and short stature which may have functional benefits [22,23]. It is well documented in our cohort and in existing literature that steroid-treated boys with DMD have short stature from early in childhood and that finding is exacerbated with increasing age [10,11,42] As obesity becomes more severe, weight may have a greater effect on functional mobility, but we were unable to stratify the obesity category further according to obesity severity due to small sample sizes.

We also identified a number of factors which were associated with increased or decreased likelihood of obesity. Seeing the neuromuscular clinic dietitian was associated with an increased risk of obesity. This is likely related to those who have already developed obesity being referred to the neuromuscular clinic dietitian for weight management advice. Scoliosis surgery was associated with a lesser risk of obesity. There may be several explanations for this finding. Steroids are associated with a higher BMI but are also protective against scoliosis; as such, those who are steroid-naïve or have been treated with steroids for a shorter period may have a lower BMI but also develop scoliosis and require surgery [8,51]. Reduction in BMI z-score following scoliosis surgery was also observed in post-hoc analysis in the current study. Individuals who undergo scoliosis surgery may be more at risk of underweight rather than obesity due to post-operative hypermetabolic state and weight loss. Complications such as infections are not uncommon after scoliosis surgery [52] and may further increase energy and protein requirements which, if not managed appropriately, can lead to malnutrition. Both extremes of BMI status (underweight and obesity) are important considerations pre- and post-operatively for scoliosis correction. In a large (*n* = 1291) cohort of young people with neuromuscular disorders (diagnoses were unspecified) who underwent posterior spinal fusions, those with obesity had a higher risk of surgical site infections, wound dehiscence, urinary tract infections and hospital readmissions [53]. Interestingly, we observed a decreased likelihood of obesity in those with neurodevelopmental disabilities. In those without DMD, neurodevelopmental disabilities are commonly associated with an increased risk of obesity [54,55]. A lower likelihood of obesity in our cohort may be due to the presence of feeding difficulties or the use of stimulant medications [55,56].

Strengths of this study include the large number of anthropometric observations collected across both ambulatory and non-ambulatory disease phases. The number of observations analysed is comparable to previous large international studies of growth in DMD [11,13]. Our cohort was predominantly steroid-treated, which is the mainstay management strategy for DMD [32]. However, due to the small number of steroid-treated boys, this meant we were not able to adjust all analyses for steroid treatment. A limitation of this study was its retrospective design, and as a result we were able to record milestones only as recorded in the medical records, rather than prospectively or systematically. The retrospective design also did not allow for standardised assessment of anthropometry. Longitudinal, prospective studies exploring the impact of BMI status on clinical milestones would overcome this issue. While we had a large sample of anthropometric observations, patient sample sizes for clinical milestone analyses were often small due to missing data. For some clinical outcomes- such as timed function tests, there may have been insufficient power to detect differences across BMI status. We analysed data for 158 individuals with DMD from the largest single paediatric neuromuscular clinic in Australia (at RCH) with an average of approximately nine years of follow-up data. Therefore, to obtain sufficient power for time-to-event analysis in future studies, data from multiple sites will be required. Larger sample sizes would also enable further exploration of the impact of minority BMI status categories such as underweight. This study was also limited by adult data not being available. Some milestones are more likely to occur in adulthood, such as FVC < 1 L which occurs early in the second decade of life [35]. Tracking patient data into adulthood through collaboration with adult neuromuscular clinics is recommended to understand the impact of BMI status on clinical outcomes occurring during the late non-ambulatory phase.

## 5. Conclusions

Obesity disproportionally affects young people with DMD compared to general populations; this can lead to earlier fractures and OSA diagnosis. The impact of obesity on physical function in DMD remains unclear. Close monitoring of growth, fracture risk factors and OSA symptoms is recommended for all young people with DMD, but especially for those with obesity. There is a need for larger, multi-site, prospective studies that track data into adulthood to understand the effect of all BMI status categories, including underweight, on clinically relevant milestones.

## Figures and Tables

**Figure 1 nutrients-14-03304-f001:**
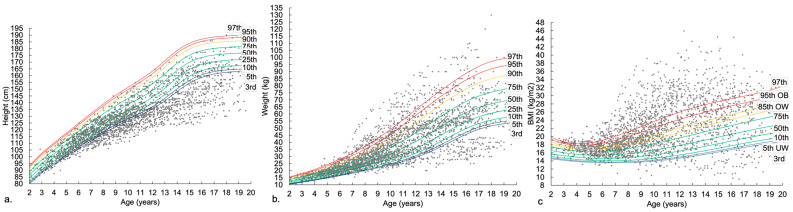
(**a**) Height (cm) (**b**) weight (kg) and (**c**) BMI (kg/m^2^) measures for DMD population overlaid on CDC growth charts.

**Figure 2 nutrients-14-03304-f002:**
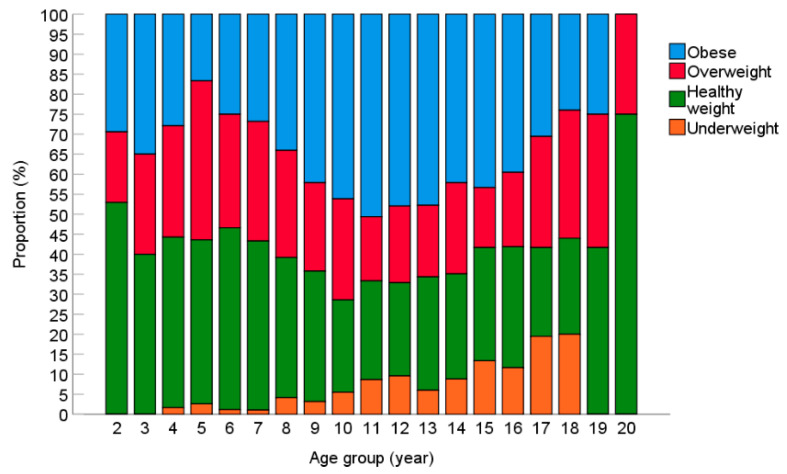
BMI Status Across Age Groups. Sample sizes for age groups: 2 (*n* = 17); 3 (*n* = 40), 4 (*n* = 61), 5 (*n* = 78), 6 (*n* = 88), 7 (*n* = 97), 8 (*n* = 97), 9 (*n* = 95), 10 (*n* = 91), 11 (*n* = 81), 12 (*n* = 73), 13 (*n* = 67), 14 (*n* = 57), 15 (*n* = 60), 16 (*n* = 43), 17 (*n* = 36), 18 (*n* = 25), 19 (*n* = 12), 20 (*n* = 4).

**Figure 3 nutrients-14-03304-f003:**
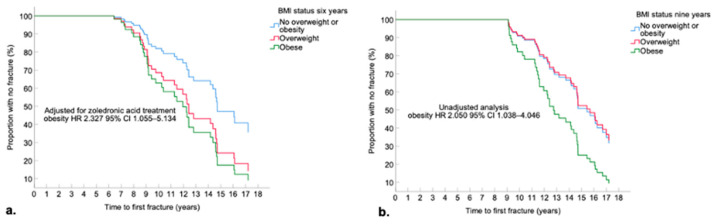
Kaplan-Meier Curve Time to First Fracture Stratified by BMI Status at Ages (**a**) Six Years (**b**) Nine Years.

**Figure 4 nutrients-14-03304-f004:**
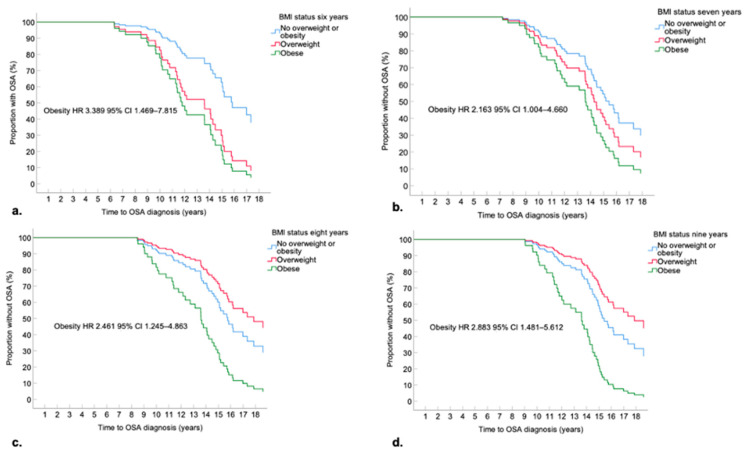
Kaplan-Meier Curve Time to OSA Diagnosis Stratified by BMI Status at Ages (**a**) Six Years (**b**) Seven Years (**c**) Eight Years (**d**) Nine Years.

**Table 1 nutrients-14-03304-t001:** Clinical Milestones Used in Time-to-Event Analyses ^1^.

Outcome Measure (Observations *n*)	Time to Clinical Milestone
Loss of ambulation (77)	First documentation of loss of ambulation
Timed 10 m walk/run (528)	First >10 s time for a 10 m walk/run [32,33], first 7–10 s time for a 10 m walk/run
Timed supine-to-stand (420)	First >30 s time to stand from supine [32,33], First >7 s time to stand from supine
Timed stair climb (470)	First >8 s time for a 4 stair climb [32,33]
NSAA (468)	First NSAA score 9 [34]
6MWD (83)	First 6MWD < 325 m [32,33]
FVC (1002)	First FVC < 1 L [35]
OSA diagnosis (71)	Polysomnography-confirmed OSA diagnosis
Nocturnal hypoventilation diagnosis (25)	Polysomnography-confirmed nocturnal hypoventilation diagnosis
CPAP commencement (26)	CPAP initiation
Bi-level commencement (22)	Bi-level initiation
Scoliosis diagnosis (48)	First documentation of xray-confirmed scoliosis
Scoliosis surgery (15)	Scoliosis corrective surgery
First fracture (crush or other) (71)	First fracture

^1^ Abbreviations: 6MWD; 6 min walk distance; CPAP; continuous positive airway pressure, FVC; forced vital capacity, NSAA; North Star Ambulatory Assessment; OSA; obstructive sleep apnoea.

**Table 2 nutrients-14-03304-t002:** Clinical Characteristics (*N* = 158).

Outcome Category	Outcome Measure ^1^	
Age and length of follow-up	Age at diagnosis (years) (*n* = 157), mean ± SD	4.2 ± 2.1
	Age at first neurologist appointment (years), mean ± SD	4.5 ± 2.5 ^2^
	Length of total follow-up (years), mean ± SD	8.7 ± 4.7 ^3^
DMD mutation type, *n* (N%)	Deletion	93 (58.9)
	Duplication or duplication/triplication	18 (11.4)
	Point mutation	29 (18.4)
	Genetic testing conducted but mutation not identified	5 (3.2)
	Genetic report not found (diagnosed with muscle biopsy)	13 (8.2)
Dystrophin isoforms maintained, *n* (N%)	Dp260, Dp140, Dp116 and Dp71	36 (22.8)
	Dp140, Dp116 and Dp71	21 (13.3)
	Dp116 and Dp71	70 (44.3)
	Dp71	5 (3.2)
	Nothing maintained	8 (5.1)
	Exons affected not available	18 (12.7)
Steroid treatment	Steroid-treated, *n* (N%)	142 (89.9) ^4^
	Age at steroid commencement (years, *n* = 139)	6.6 ± 2.3
	Prednisolone only, *n* (N%)	81 (51.3)
	Prednisolone then deflazacort, *n* (N%)	58 (36.7)
	Other corticosteroid, *n* (N%)	3 (1.9) ^5^
	Steroid-naive, *n* (N%)	16 (10.1)
Anthropometric measures (*n* = 156)	Height z-score (observations *n* = 2480), mean ± SD [range]	−1.47 ± 1.40 [−6.06–2.97]
	Weight z-score (observations *n* = 2902), mean ± SD [range]	0.10 ± 1.56 [−8.51–3.24]
	BMI observations per individual over total follow-up	16 ± 10
	BMI observations per individual per year of follow-up	2 ± 1
	BMI z-score (observations *n* = 2456), mean ± SD [range]	1.00 ± 1.60 [−14.23–3.36]
	Change in BMI z-score across total follow-up (*n* = 150), mean ± SD [range]	0.23 ± 1.58 [−7.36–4.59]
Co-morbidities, *n* (N%)	Any neurodevelopmental disability diagnosis	39 (24.7) ^6^
	Any mental health diagnosis	22 (13.9)
Deaths	Age at death (*n* = 10), mean ± SD	15.7 ± 3.0

^1^ Missing data is as follows: age of diagnosis (*n* = 1), mutation type (*n* = 1, genetic testing performed but mutation type not available), steroid commencement date and age (*n* = 3), age at loss of ambulation (*n* = 5). ^2^ Mean age at diagnosis is lower than the mean age at first neurologist appointment due to antenatal diagnoses and patients being diagnosed at external clinics (interstate or overseas) before transferring care to RCH. ^3^ Length of follow-up: date of first neurologist appointment to last neurologist/neuromuscular appointment. ^4^ Steroid-treated: treated with steroids for any length of time during the follow-up period. ^5^ Three children were involved in a clinical trial of an alternate corticosteroid. ^6^ Any neurodevelopmental disability diagnosis: autism spectrum disorder, intellectual disability, attention deficit hyperactivity disorder or other developmental disorders.

**Table 3 nutrients-14-03304-t003:** Descriptive Data for Clinical Milestones ^1^.

Milestone Type	Clinical Milestone	Event Occurred during Follow-Up, *n* (%)	Time to Event (Age in Years) Median (IQR)
Physical function	Non-ambulant	82 (51.9)	11.0 (9.3, 12.4), *n* = 77
	10 m walk/run in 7–10 s	56 (35.4)	9.7 (8.2, 11.0)
	10 m walk/run in >10 s	35 (22.2)	10.8 (9.5, 12.0)
	Supine-to-stand in >7 s	58 (36.7)	9.5 (8.3, 10.6)
	Four stair climb in >8 s	45 (28.5)	10.2 (8.4, 11.6)
	NSAA total score 9	31 (19.6)	11.6 (9.4, 13.1)
	6MWD < 325 m	17 (10.8)	11.6 (10.5, 13.6)
Respiratory function	OSA diagnosed	72 (45.6)	13.1 (9.7, 15.0), *n* = 71
	Nocturnal hypoventilation diagnosed	27 (17.1)	15.3 (13.3, 16.5), *n* = 25
	CPAP initiated	24 (15.2)	13.7 (12.5, 16.0)
	Bi-level initiated	20 (12.7)	15.5 (14.0, 16.5)
	Both CPAP and Bi-level initiated	2 (1.3)	-
	FVC < 1 L on lung function test	27 (17.1)	13.8 (11.2, 16.4)
Fractures	≥1 fracture	71 (44.9)	11.4 (8.8, 14.1)

^1^ Missing event dates: loss of ambulation *n* = 5, OSA diagnosis *n* = 1, nocturnal hypoventilation diagnosis *n* = 2 Abbreviations: 6 MWD, 6-min walk distance; CPAP, continuous positive airway pressure; FVC, forced vital capacity; OSA, obstructive sleep apnoea.

## Data Availability

Not applicable.

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
