# Peer review of "The Relationship between Obesity and Clinical Outcomes in Young People with Duchenne Muscular Dystrophy"

_nutrients, 2022, doi:10.3390/nu14163304_

Round 1
Reviewer 1 Report
This paper analyse the correlation between overweight/obesity and fractures, physical function milestones and obstructive sleep apnea (OSA) in Duchenne Muscular Dystrophy (DMD) patients.
The authors report a retrospective analysis of 158 DMD patients and showed a positive correlation between obesity and fractures or OSA, while no correlation between obesity and physical functional milestones. They conclude that early weight management is important to prevent fractures and OSA.
The paper is interesting and well written; however, I have some doubts on the data analysis and conclusion.
While there is a consensus on the correlation between obesity and OSA, the risk of fractures in DMD patients is demonstrated depend principally by steroid-treatment, because it is a common side effect of this treatment.
In fact, the same authors in the paragraph “Factors Associated with Obesity” demonstrated that overweight/obesity in their patients is associated to steroid-treatment.
To demonstrate the role of obesity, independently by steroid-treatment, is necessary analysed the correlation of fracture in steroid-treatment group vs. no steroid-treatment.
Moreover, there are two subgroup in the steroid-treatment group: prednisolone and mixed (prednisolone then deflazacort). It could be interesting analysing the correlation between prednisolone vs. mixed-treatment and fractures.
Minor Revision: change “Between” with “between” in the title.
Author Response
Dear Reviewer 1,
Thank you for your consideration of our manuscript.
Please see below response to your suggestions.
Kind regards,
Authors
While there is a consensus on the correlation between obesity and OSA, the risk of fractures in DMD patients is demonstrated depend principally by steroid-treatment, because it is a common side effect of this treatment.
In fact, the same authors in the paragraph “Factors Associated with Obesity” demonstrated that overweight/obesity in their patients is associated to steroid-treatment.
To demonstrate the role of obesity, independently by steroid-treatment, is necessary analysed the correlation of fracture in steroid-treatment group vs. no steroid-treatment.
We have now adjusted the fracture analysis by steroid-treated vs. naïve. Please see section “Obesity and First Fracture”. We also included an interpretation of this in the discussion.
Moreover, there are two subgroup in the steroid-treatment group: prednisolone and mixed (prednisolone then deflazacort). It could be interesting analysing the correlation between prednisolone vs. mixed-treatment and fractures.
Due to the small sample size for the Cox regression analyses, we do not think we are powered to analyse by steroid subgroup. Furthermore, recent research has highlighted there may be no differences in fracture rate between steroid types. We have now revised the paragraph in the discussion as follows:
“Obesity at two time points, six and nine years, increased the likelihood of sustaining a fracture earlier. These significant findings were independent of steroid treatment in adjusted analyses. Analysis at other ages may have been limited by small sample sizes. Increased risk of fractures in children with obesity has also been observed in typically developing populations [40]. In DMD, both steroids and weight gain may lead to fractures in DMD. Steroid treatment can cause weight gain which may independently increase fracture risk [8] and steroids can also impact bone mineral density and lead to fractures [41]. Our sample size was too small to analyse differences between steroid types (e.g. deflazacort vs. prednisolone), however in a recent randomised clinical trial there were no significant differences in rates of fractures between daily prednisolone and deflazacort [42]. In those with obesity and/or those taking steroids, there is a need for close surveillance of fracture risk factors such as vitamin D deficiency, low dietary calcium intake and falls. Dietitians and physiotherapists play an integral role in advising families on how to mitigate these risk factors to help prevent fractures, which cause significant burden and can lead to loss of ambulation and decline in function and strength [43].”
Minor Revision: change “Between” with “between” in the title.
Changed.
Reviewer 2 Report
I consider the theme of this article of scientific interest. The data are well presented, but they are influenced by medication. Most of the patients were treated with steroids, obesity and fracture being probably in relation with this therapy, too. The charts from fig. 1 are very small and difficult to observe.
1. Title – The title is suggestive enough for the topic of the article.
2. The abstract is informative enough.
3. Keywords – adequate.
4. The introduction presents the data published until now in relation with the article’s main topic in a clear and concise form.
5. Material and method: The main limitation of the study is represented by retrospective analysis of patients’ records, even for anthropometric measurements, but I consider that the influence could not be very significant. The measurements of Height and Weight were done by a medical qualified person. The number of cases included in this study is quite impressive for such a disease – 158 young patients monitored in a specialized center, over 90% of them having a genetic test for the disease. Statistical analysis is well described, well done and, probably, is the most important strength of this publication. The data were analyzed with a lot of statistical method and models to find something significant.
6. Results – are well presented, but they are influenced by medication. Most of the patients were treated with steroids, obesity and fracture being probably in relation with this therapy, too. The charts from fig. 1 are very small and difficult to observe.
7. The discussions address the subject of the paper.
8. Conclusions of this study are well selected.
9. The references are written according to the instructions for authors.
10. The English language is good, in my opinion, maybe some minor grammar mistake should be corrected.
Author Response
Dear Reviewer 2,
Thank you for your consideration of our manuscript.
Please see below response to your suggestions.
Kind regards,
Authors
I consider the theme of this article of scientific interest. The data are well presented, but they are influenced by medication. Most of the patients were treated with steroids, obesity and fracture being probably in relation with this therapy, too. The charts from fig. 1 are very small and difficult to observe.
Thank you, the font size of figure 1 has now been increased.
Round 2
Reviewer 1 Report
Dear Authors
I have suggested to analyze the correlation of fracture in the steroid-treatment group vs. no steroid-treatment. You have answered that it was made in the section “Obesity and First Fracture”, however, you have adjusted the fracture analysis considering the zoledronic acid, a bisphosphonate not a steroid.
Kind Regards
Author Response
Dear Reviewer,
Thank you for your ongoing review of our manuscript. Regarding your comment, we have clearly stated our adjustment for steroid status. Please see details below for the line numbers where this has been described.
In section 3.4.2 Obesity and First Fracture
Line 240-241 Obesity at six years was also associated with younger age at first fracture when adjusted for steroid status prior to first fracture (obesity HR: 2.196; 95% CI 1.019-4.733).
Line 243-244 This remained significant ... and when adjusted for steroid status (obesity HR: 2.169 95% CI 1.076-4.373).
Line 249-250 In unadjusted and adjusted analyses, obesity at five, seven and 249 eight years did not impact time to first fracture (see supplementary data).
Round 3
Reviewer 1 Report
Dear Authors,
the manuscript is now improved. I suggest inserting in the discussion that the few numbers of subjects steroid naïve are a limitation of the study.
Kind regards
Author Response
Dear Reviewer,
Thank you for your ongoing review. We have stated the small number of steroid-treated participants as a limitation on line 368.
Thank you